# Adolescence as the Context for Understanding Young Mothers’ Engagement with Health Promotion: A Phenomenological Exploration [note 1]

**DOI:** 10.3390/children10050904

**Published:** 2023-05-20

**Authors:** Catherine Ellis, Peter Sidebotham

**Affiliations:** 1Department of Nursing, Midwifery and Health, Faculty of Health and Life Sciences, Northumbria University, Newcastle upon Tyne NE7 7XA, UK; 2Warwick Medical School, University of Warwick, Coventry CV4 7AL, UK

**Keywords:** adolescent mother, young mother, parenting behaviour, decision making, parenting practices, IPA: interpretative phenomenological analysis

## Abstract

Background: The current social construction of young mothers is generally negative, pointing to a lack of engagement with universal services and poor outcomes for their infants and children. However, qualitative studies offer an alternative, more positive construct of young motherhood. Understanding the context of young motherhood can improve the relevance and efficacy of health promotion directed to this group of high-risk mothers. Aim: To explore the lived experience of young women transitioning to motherhood to better understand their experiences and perspective; and what influences their engagement with health promotion aimed to support safer parenting practices and whether their behaviour changes over time with exposure to parenting health promotion. Method: Longitudinal Interpretative Phenomenological Analysis (IPA) was used with five first-time mothers identified with characteristics known to influence poorer outcomes for infants and children such as low educational achievement and economic disadvantage. Participants aged 16 to 19 years were recruited antenatally. Serial in-depth interviews were conducted at three time points during the ante- and post-natal periods. Interviews were transcribed and data were analysed inductively following the prescribed method of double hermeneutic analysis for IPA. Finding: Three themes were identified from the full study: Transition, Information, and Fractured application; the focus of this paper is Transition. Transition revealed that becoming mothers impacted key adolescent developmental tasks; their identity and relationships were significantly affected, both positively and negatively and adolescent brain development influenced behaviour and decision making capability. Adolescence influenced how these young mothers engaged with and interpreted parenting health promotion messages. Conclusions: Young mothers in this study operate within the context of adolescence. Adolescence impacts participants’ decision making activity and early parenting behaviours which informs the debate on why young mothers may fail to reduce risks for their infants. This insight can contribute to the development of more effective health promotion/educational strategies, and support professionals to better engage with this high-risk group to improve early parenting behaviour and subsequently improve outcomes for their infants and children.

## 1. Background

The general perception of young mothers in our society and culture is often negative; teenage parents are typically portrayed as deviant, unproductive, promiscuous, irresponsible, and lacking aspiration [1,2]. As a social construction, mothering is portrayed as a demanding task performed by appropriately aged and competent women. Within this construct, teenagers do not fit with the ‘good mother’ image and are viewed as a social problem and unprepared for the parental role, with poor outcomes noted for their infants and children [2,3,4,5].

Within this, infant mortality is identified as a key metric, including mortality from Sudden Infant Death Syndrome (SIDS). In England and Wales, around 200 infants die each year from SIDS [6] and families living in socio-economic disadvantage are at increased risk, with young white British mothers aged 16 to 20 disproportionately represented. Epidemiological research has identified modifiable risk factors for SIDS, for example, prone and side sleep position, soft bedding, paraphernalia in the cot, overheating of the infant, hazardous co-sleeping, and smoke exposure, suggesting that many of these deaths are preventable [6,7,8,9,10]. Research has identified that young mothers are more likely to expose their infants to increased risks in the sleep environment and have less knowledge of risk reduction strategies than mothers considered to be low risk [11,12,13], despite the availability of health education programmes [9,14].

While there are undoubtedly poorer outcomes for some young parents and their children, much of the evidence to support teenage pregnancy and parenting policy originates from quantitative research, which fails to consider or explain the association between being young parents and these poorer outcomes [15,16,17,18]. Qualitative perspectives that include the voice of the young mother present an alternative construction. Early motherhood has been described as a landmark event, linked positively to a teenager’s development and transition to adulthood, bringing a sense of responsibility, purpose, and motivation to change behaviour [1,4,5,16,17,18,19,20,21,22,23]. While identifying the positives, young mothers were acutely aware of the negative stereotypes [5,16,24,25]. They reported experiencing high levels of criticism from both the public and professionals of their choices, parenting skills, knowledge and ability, and felt that their parenting had to be of a higher standard than for older mothers [26,27]. Young parents must be seen to be ‘perfect parents’ while coping with limited resources and support [19,28]. Young mothers also described feeling ‘not permitted’ to enjoy their pregnancy or planning for motherhood, and while most pregnant young women enjoyed the physical presentation of pregnancy, for some, the ‘baby bump’ presented a visible target to be judged by the public [29]. In some communities, early parenthood is the norm and is valued and accepted as a positive choice [3]. However, living at home has implications for young mothers and can have a significant influence on their decision making and parenting practices [30,31]. While the literature supports that most young mothers are well able to make their own decisions and offer sound rationale for these, reduced personal resources and external factors influence their decision making process and can have a significant impact on their ability to parent.

The public health outcomes framework for England for 2013–2016, set out in the Department of Health’s publication “Healthy lives, healthy people: Improving outcomes and supporting transparency” [32], provided a framework for health services to measure and improve health outcomes for all sections of society. The framework indicators aim to reduce low birth weight babies, promote breastfeeding, reduce maternal smoking, conceptions to under 18 –year –olds, increase population vaccine coverage, and reduce infant mortality. While service provision is required to achieve these goals, the primary responsibility falls to the individual to modify their behaviour to achieve improvement in these outcomes. There are several programmes of work to support maternal and child health service delivery and achievement of these outcomes; the healthy child programme [33], the Family Nurse Partnership [34,35], the Implementation Plan to reduce health inequalities and infant mortality [36], and relevant guidance on smoking, obesity, and pregnancy from the National Institute for Health and Care Excellence [37]. However, the provision of these universal and targeted services has a substantial resource requirement; financial austerity, combined with the growing burden of adult social care, means that funds are stretched across competing priorities, and review, consolidation and cuts to services have the greatest impact on the most disadvantaged groups in society. It cannot be ignored that the effects of living in poverty and experiencing social marginalisation are likely to have a significant impact on young mothers and provides the basis of the cultural context of their lifestyle and parenting choices. Having identified the risk factors that increase the risk for poorer outcomes within families, it is essential to recognise the risks that can be modified and design health promotion and education strategies that are effective and relevant for this group of parents.

Engagement with professionals whose attitudes are likely to be embedded in the middle-class norm can alienate young mothers and prevent them from accessing services which can impact their health, wellbeing, and their experience of early motherhood. Low self-esteem, lack of confidence, and a perceived lack of ability to be a good mother can undermine young women’s experiences and could be different if the negative stereotype was less dominant [5]; and young mothers experience further criticism when they do not engage with services from which they feel alienated [3,5,26,31]. While the majority of young mothers adapt well to the role of parenting, some struggle to cope and may feel alienated from service provision or may reject support perceived as authoritarian and feel they lack control over their own lives; for these young women, the transition to motherhood is challenging.

Understanding how young mothers prepare and transition to motherhood, what influences their decision making and parenting behaviours and how health services support this was the subject of a wider study to explore young mothers’ understanding of safe-sleep and infant-care practices that increase the risk for SIDS, and how they make decisions and apply knowledge of risk factors in the infant sleep environment. This paper focuses on the first key theme of Transition, which identified the context for young mothers’ engagement with health promotion, and their knowledge and application of current safer parenting recommendations.

## 2. Method

A qualitative and longitudinal approach using Interpretative Phenomenological Analysis (IPA) [38] was selected to explore the lived experience of a homogeneous group of first-time young mothers. Identification and recruitment of participants were facilitated by a specialist teenage pregnancy partnership. Five participants who met the inclusion criteria (white British, aged 16–20, first baby, and economic disadvantage) were recruited. Data were gathered by serial in-depth interviews at three time points: 32–36 weeks antenatally (I/V 1) and four (I/V 2) and 16 (I/V 3) weeks postnatally to reveal parenting practices and decision-making processes, and how these might change over time. Interviews were based on three topic guides of six open questions covering current personal circumstances and lifestyle; physical and emotional considerations of becoming a mother; exposure to and understanding of relevant health promotion messages; other sources of information and support; and intentions with regard to parenting practices and their foundation. Questions were open and invited respondents to expand on their perspectives rather than to ask specific questions regarding current recommendations. This approach aimed to identify actual knowledge and application of safer parenting practices and avoid leading participants to respond to items raised by the interviewer, as required by the IPA approach. All interviews were face-to-face in participants’ homes, audio-recorded, and transcribed. The post-natal interviews afforded the opportunity to consider both the infant’s sleep environment and the interaction between mother and infant. Participants were each assigned a pseudonym to protect confidentiality and enhance reporting fluency. Participant profiles are presented in Table 1. Ethical approval for the study was granted: Ref:16/NS/0087.

The study was conducted in a large English city with significant deprivation. Approximately 45% of births in the locality occur in the most deprived quintile of the population with a birth rate of 18.2 per 1000 for women under 20 years of age, in comparison to the UK rate of 15.6 [39]. The local NHS Trust employed two specialist teenage pregnancy midwives based at a Children’s Centre in one of these most deprived areas, highlighting the need for specialist service provision in the city.

IPA methodology aims to explore participants’ views and understand how they make sense of the phenomenon of interest. The researcher then interprets the meaning participants ascribe to the phenomenon, to provide a technical account of the phenomenon to enhance wider understanding [40]. The key principal of IPA is the idiographic narrative; to move from the participant’s idiographic account through interpretation to the researcher’s technical account requires that data be subject to a six-step analysis process: reading the transcript, notation, developing themes, searching for connections, moving to the next case, and repeating stages one to four, then identifying patterns across all cases [38].

Interviews were transcribed verbatim; using a hard copy of the transcript, initial noting and highlighting captured expressions and comments of note, the language used, reference to an issue or idea, or narrative that generated a question; these were recorded on coloured ‘post it’ notes. The resultant ‘post its’ were transferred to flip charts and clustered. Flip chart clusters and annotated transcripts were then reviewed; clusters were amalgamated, subsumed, or separated to create new clusters. This process revealed possible emergent theme groupings which were tabulated and captured into a ‘concept table’ for each interview, then collated across all interviews and time points individually before collating as a whole.

Due to the complexity of analysing longitudinal data, each interview was treated as a discrete case study before moving to the next. This approach was used to preserve the idiographic focus of the analysis phase and to prevent cross-contamination of themes from one interview to another, and between participants, before moving to analyse themes across the group. To strengthen quality and audit, Yardley’s [41] framework for assessing quality in qualitative research, which is considered particularly useful for application to IPA, was applied to this research; and an audit process of cross checking with a colleague independent of the research, but familiar with the IPA analysis process, was conducted to identify similarity in emergent themes on three randomly selected clean transcripts. A high degree of similarity was identified between researcher and auditor, supporting transparency in the interpretation of the data. 

The research question aimed to understand the lived experience of young women through the process of pregnancy and becoming mothers in order to understand how they received, interpreted, and applied safer sleep information. The superordinate theme of Transition (subject of this paper) captured the concept that adolescence was the context for understanding how information shared with participants was received and processed (theme two: information), and ultimately how that information was translated and applied into the infant sleep environment and to other parenting practices (theme three: fractured application) (Figure 1).

Adolescence and the journey to motherhood are a priori, both transitional; however, specific subthemes identified in the data revealed that transition emerged as an appropriate superordinate theme. While aspects of receiving information and of translating and applying that information were reflected in the other superordinate themes of information and fractured application, they were influenced by transition.

This is the first of three inter-related papers, each presenting one of the three superordinate themes from the wider study. Transition, as the context for young motherhood, is presented here.

## 3. Findings

Several highly complex changes were taking place concurrently in the lives of these young women, not only were they transitioning between the worlds of adolescence and adulthood, but they were also negotiating the unexpected journey to motherhood. Almost all the themes emerging from the data were shared by all participants, albeit from slightly different perspectives. Transition was identified as the superordinate theme, with nested concepts of *identity*, *relationships*, and *adolescent brain development* identifiable through several subthemes of *growing up*, *becoming a mum*, and *relationships* (Figure 2). Each of these three subthemes will be explored in turn, drawing out the data informing that theme and how they relate to the overall superordinate theme of transition.

Acceptance of pregnancy was minimal for four participants; Evie admitted to wanting a baby at this time but Mia and Sophia both expressed wishing they had waited until they were older. All were financially and practically supported by their parent/s or extended family. While all had been in receipt of specialist and enhanced services antenatally, only two were eligible for extra support in the post-natal period. All had a variable history of engagement with maternity and health visiting services, both pre and post birth.

### 3.1. Growing Up

The transition to young adulthood can take many years to complete, often continuing beyond mid-twenties [42]; however, the transition to motherhood is a time-limited process. All participants demonstrated a reluctance to move toward adult independence but wanted to be considered an adult regarding becoming a mother. The antenatal data highlighted that there was a ‘disconnect’ between the adolescent self and the pregnant self; all participants had difficulty visualising the reality of being a mother and were not or did not want to think that far ahead; the actual embodiment of pregnancy was ‘unreal’ for them. They were ‘just pregnant’, and their focus was on what was relevant and important in their lives at that point; studying for exams, focusing on their health, continuing to work, or finding somewhere to live. In response to questions about their pregnancy and preparation for motherhood, they were required to acknowledge their pregnancy as a reality. In doing so, participants presented an imagined future, consisting of a supportive partner, a home, and the expectation that they would be living as a family, despite only two of the participants stating they were in a relationship with the father. All participants except Grace were living in their family home; however, while Grace was given a council flat, she never moved in. Participants’ parents continued to provide support, both financially and practically; so, although they were transitioning to motherhood, these young women were still living and being treated as teenagers:


*‘I like the independence of having a flat. Yeah, I’d like my mum to move in.’*
(Grace, 17: I/V 1)


*‘I’m living here with my mum, dad and my brother. So, got all the help, but I wouldn’t move out… I’m glad I’ve got my mum there, so I know she’ll help me, you know she’s had four children, so she like knows what she’s doing, and I think I need that help.’*
(Evie, 19: I/V 1)

While participants wanted to be seen to be growing up and treated differently, they expected that their parents would continue to look after them, provide for them, and help with infant care. Interestingly, during the interviews with participants, often their parent/s were present and offered their perspective, revealing that they were equally complicit in maintaining their daughters’ dependence and were less likely to view their daughters as moving toward adulthood.

Development of identity is a key stage of adolescence which is supported by social interaction and relationships. Younger participants, in particular, experienced disruption in the development of their identities. Mia and Sophia met at their new school, then met other pregnant young women and mothers; however, this network was small. These new friendships, born from necessity, were not viewed as ‘real’ friendships, and the younger participants distanced themselves from their previous peer groups, also refraining from interactions with them on social media. However, the older participants also expressed feeling distanced from their peers, although they appeared to accept and manage this better.


*‘that group’s kind of changed a lot, cos it’s like they seem really immature compared to me now. Like, they just wanna go out all the time and like, ‘oh, I’m going to so and so’s, and I’m gonna like get drunk’. ‘Oh, that’s great, have fun, you wreck your liver!’ Laughs. You know?’*
(Sophia, 16: I/V 2)


*‘They’ve been round and they are really supportive, but it’s kind of hard for me to…, I feel we’re just not on the same level anymore, I’m focused on her [baby] and they…, just because they haven’t got children and stuff they just, they want to go out, and it’s just different now, so yeah. And you just get on with your life, don’t you really?’*
(Ruby, 18: I/V 2)

Acquiring skills in decision making is an essential aspect of growing up and moving to adulthood. These young women were being accelerated through this important developmental stage and as such their decision making skills vacillated between impulsive adolescent processes and reasoned and mature adult decision making. There was evidence of rational decision making regarding quitting smoking and the selection of infant feeding methods for example. However, many of the decisions they made appeared either impulsive, lacked explicit rationale, or were supported by flawed or incomplete knowledge; occasionally participants were just unable to identify why they were making decisions or were ‘instructed’ by someone they trusted and did not question or challenge this:


*[Where will she sleep?] Next to my bed [how did you decide that?] Just thought of it’*
(Grace, 17: I/V 1)


*‘I just liked it, so I bought it [cot], I didn’t look into anything like Google. My auntie’s [aged 25] just had a baby, and she like Googles everything, it’s gotta be like perfect detail of what it does and why this is going to be good… but I didn’t do that, I just, I liked it, so I just bought it.’*
(Ruby, 18: I/V 1)


*[Is there anything that you’ll change about the Moses basket?] ‘No, oh… the mattress [laughs]’ [why?] ‘Cos I have to…’. ‘cos I’ve been told to [laughs] by the midwife. Erm, well she said a new baby needs a new mattress’ [why was that?] ‘erm, not sure.’*
(Mia, 16: I/V 1)

### 3.2. Becoming a Mum

The ‘disconnect’ between adolescent self and self as mother continued through birth and the early days of motherhood. Participants expressed feeling completely overwhelmed with the focus on ‘survival’ and the present. Language such as ‘unreal’, ‘she can’t be mine’, and ‘detached’ was evident in accounts of their initial feelings of having a baby. However, as participants moved beyond the initial weeks and considered what being a mother was, alongside their current reality, they were much more able to share their experiences of their lives as mothers. They were more aspirational and talked of responsibility and having to grow up. The presence of the infant focused their attention, as in these two examples:


*‘…it sort of gives me motivation, to study and to get, you know, go far, go further in life.’*
(Sophia, 16: I/V 2)


*‘Yeah, I’m doing a dual accountancy apprenticeship, so yes, definitely I’m going back to that. So… I want to get it all done now quickly for her. [does having a baby motivate you?] Yeah, 100%. I don’t even think of me now, I just think of her’*
(Ruby, 18: I/V 2)

The role of mother was divided into the ‘work of mothering’ and ‘validation as mother’ (changing identity). The work of caring for an infant was described as hard, tiring, relentless, frustrating, and boring. Sophia and Mia, both 16, expressed feeling ‘trapped’ in their bedrooms with a baby all day and were more likely to talk in terms of the hard work of motherhood. By contrast, Evie and Ruby, both 19, viewed the transition to motherhood as easier than expected and wanted to be seen as competent mothers.


*‘It’s a bit stressful, it is a lot of hard work, and at times you’ll be really cranky [laughs]. Some days it’s like, some days, I have no idea. He’s like, such a crier at night and it’s like I don’t know what to do, I’ve done this, I’ve done that, I’ve done everything, what do I do? But, it’s just he wants to be held.’*
(Sophia, 16: I/V 2)


*‘And then, I like, clean up all his nappies from the night, and then wash his bottles and normal stuff, all the boring stuff. Like, changing his nappy. And then don’t know really, I just find everything boring’ [laughs]. [why is that?] ‘Erm… like ‘cos he can’t speak to you and like, he cries’ and you’re like ‘I don’t know what to do, tell me what you want’ and you have to figure it out and it’s pretty boring.’*
(Mia, 16: I/V 2)


*‘she has had a couple of days when she can’t make out what she wants, she can’t make out whether she wants food, a nappy change or just a bit of love, so it can take a little longer for that.’*
(Evie, 19: I/V 2)

Evie’s perspective is that when her baby settles easily, it is down to her mothering skills, however, when she cannot settle, the infant is to blame, which might illustrate the tension in moving from the worlds of adolescence to those of adulthood and motherhood.

### 3.3. Relationships

Validation as a good mother was important to all participants and could be identified from how they described relationships, interactions with professionals, and the public scrutiny they experienced. Relationships with peers was often mentioned. Mia and Sophia both described their friends as having changed, which is an interesting perspective. Mia stated she was ‘becoming boring’ and that she must live by ‘mum rules’ now, while Sophia was disparaging about the ‘antics’ of her old friends. However, within their new ‘mum friends’ group, Sophia was validated by the others for her knowledge and experience of motherhood, which she found positive and reinforcing. While they both felt they were missing out on what their old friends were getting up to and felt excluded and isolated, motherhood brought some rewards; managing to settle a crying baby and smiles at bath time were related with pride.


*‘Erm… cos I don’t go out as much and I’m not as fun… laughs. I’m a mum, I’ve got mum rules’*
(Mia, 16: I/V 2)


*‘I’m the one with the oldest baby, so it’s like questions everywhere [Laughs]. I’m like the agony aunt of the group [laughs]. [how does that make your feel?] Makes me feel well proud of myself [laughs].’*
(Sophia, 16: I/V 3)

As the youngest participants, Sophia and Mia appear to have had to grow up more quickly; they have become mothers and are aware of their responsibilities in contrast to the fathers of their infants, who were able to walk away from this ‘scary’ responsibility. Sophia and Mia did not have a choice to walk away; they faced up to their responsibilities and felt proud to be mothers. However, Mia also expressed that having a baby ‘made her grow up too much’, suggesting that she wasn’t ready for such a big step towards adulthood at 16 years old, and in having to grow up, she has ‘lost’ whom she was becoming. Having a baby for Mia had a significant cost, with the resultant changes isolating her from her old friends and taking its toll on relationships both with her family and the baby’s father.


*‘I like being a mum, but… I don’t regret it, but I wish I’d had him later. Where I’ve had like a job and obviously like got my own little house and have a little family, [pause] but I don’t really regret it, at all.’*
(Mia, 16: I/V 3)

While Evie and Ruby had been working, had smaller but perhaps better-established friendships, and larger families providing stability in their relationships, they also reported changes to those relationships. They also appeared to adapt to motherhood more easily which might have reduced feelings of being ‘stuck’ or ‘trapped’, and both had ongoing relationships with their baby’s father.

Linked to validation was others’ perception of them as young mothers. Undertones of disapproval were a feature of these young mother’s experience of not being worthy, and feelings that both professionals, the public, and occasionally, their family expected them to fail at this important role, as expressed by Mia:


*‘And like some people, when you’re on the bus or something, I feel like people like, look at you, you get that look, and they’re like, they are judgemental, I’m like, I’ve took on being a mum and how can you judge me, for being, like doing that? Which I just don’t understand it.’*
(Mia, 16: I/V 2)

Sophia and Mia did not ‘fit’ into their previous peer groups, nor did they feel that they belonged accessing services for older mothers, expressing they felt judged by other mothers for being too young. While participants reported good relationships with their midwives and family nurses, shortly after birth, professional contacts receded. Evie and Ruby were happier to be left to their own devices and stated they would access services as required; however, the younger participants expressed feeling isolated even though they were receiving care via the Family Nurse Partnership (FNP). This illustrates an important point about engagement with services and the opportunity for ongoing health promotion/education from professionals; both younger and older participants identified different reasons for non-engagement with services, but all shared feelings of being judged and services not being relevant for them.

While all participants could identify at least one reason to breastfeed and all expressed in the ante-natal interview that they intended to breastfeed, participants’ experiences of being supported and advised by professionals during hospital stays were varied. Comments around support with breastfeeding revealed that midwives perhaps did not expect the young mother to persevere, and perhaps did not see the value in investing time to support breastfeeding, or perhaps the young mother did not have the tenacity to persevere and this was identified by the professional. Further exploration with participants of the support to continue to breastfeed was either circumvented or dismissed, or they claimed professionals advised them to give up. Perception of help from the midwives may have been distorted, or the reality may have been that the midwives did persuade them to bottle feed, assessing that their attitude was not one of persisting with the ‘work’ of breastfeeding. Participants could logically identify why they should breastfeed and expressed antenatally that they wanted to; however, the reality and difficulty of breastfeeding may have caused them to revert to adolescent decision making processes, leading them to abandon breastfeeding, or the belief that they could better ‘demonstrate’ they were good mothers if they bottle fed. These quotes demonstrate how quickly participants gave up breastfeeding, and although Evie reported that she breastfed for over two weeks, the reality was that she expressed her breastmilk and bottle fed her infant; so her claim that her milk supply was the issue was physiologically flawed and her perception altered to support her worldview:


*‘So, I breastfed, erm… she didn’t like it… AT ALL, [how long did you?] Two and a half weeks, she hated it. She just prefers the bottle. But there was obviously nothing I could do about that, it [milk supply] just completely cut out [what did the midwife say?] So, they said you can’t really reverse it once it cuts out so there was nothing I could do’*
(Evie, 19: I/V 2)


*‘He won’t latch on. I did try and at first, he did, but it just wouldn’t work, so… [Did you have help?] Yeah. Erm… they said if I wanted to try and do breastfeeding, they’d have to keep me in more, and I was like, no! I’m just going to bottle feed.’*
(Sophia, 16: I/V 2)


*‘Yeah, I was just adamant that I was going to breastfeed, but when she come out and she was hungry, I just said give her a bottle, and I couldn’t do it. [did you have help from the midwives?] Yeah, they helped but, I just, she was just so hungry, I just said just give her a bottle, I can try [breastfeeding] later’*
(Ruby, 18: I/V 2)

The findings present elements of the lived experience shared by these young women through the process of pregnancy and becoming mothers. Emergent and subordinate themes identified that characteristics of adolescent functioning were recurring and as such, influenced how they approached the transition to motherhood. This important and complex developmental stage provides the context for engaging with these young women and understanding what influences their decision making processes and behaviour.

## 4. Discussion

Primarily, these young women were adolescents, superimposed with their unexpected journey to motherhood. Their accounts reveal initial emotional turmoil and disconnection which was followed by a future of purpose, aspiration, and taking on the role of motherhood with all that that entailed; these positive and negative aspects of parenthood have been identified in the literature.

This study used a longitudinal approach to explore experiences and parenting practices over time. These findings extend our understanding of young motherhood by revealing that in becoming mothers there was not an automatic or accelerated transition to becoming an adult, when perhaps participants and those around them may have viewed this as the case. Data revealed examples of participants’ behaviour and decision-making processes as linked to aspects of their developmental stage, identifying the influence of adolescence on decision making and behaviour, influencing their early parenting approach. This was important to draw out as a theme, as understanding what influences young mothers’ engagement with health promotion to inform early parenting behaviour and decision making impacts infant care and potentially infant safety.

The data also revealed the impact of becoming a mother on the critical developmental stage of adolescence. Adolescence is characterised by both physiological change and psychosocial transition and progression toward taking on adult roles and independence [42,43,44], and several theorists locate adolescence as a critical developmental period in the lifespan [45,46,47]. This developmental stage requires several key tasks to be achieved which include increased decision making and abstract thinking skills (risk assessment; cause and effect), developing a sense of connectedness through peer relationships, the search for self and establishment of identity, and the move towards independence [48]. From the data, it was evident that becoming a mother, for some of the participants, had derailed some of these critical tasks.

Recent research has identified that the transition to adulthood may not be reached until the late twenties [42,43,44]. This extension of adolescence was evident within my data, demonstrated by the tension between wanting and needing to grow up, and wanting to be validated as a mother and responsible adult while still being cared for by family and continuing with their adolescent life. The new grandparents were equally complicit in encouraging or maintaining ‘dependence’ in their daughters and some professionals may have provided care and advice based on assumptions either that the young mother was processing information as an adult or they were unable to make rational decisions as they were considered too young. While this social shift of extended adolescence has been documented in the child development literature [44,49], the impact of this on adolescents as parents, was absent. Therefore, this social shift may be important in terms of its influence on aspects of adolescent parenting practices of the future. These findings contribute valuably to enhancing our understanding of the context for young mothers to understand their needs for service provision better, and consequently, more effective engagement.

Adolescence is about moving toward independence [50], and while family can be a source of valuable support, this can be experienced as overbearing and undermining for the young mother, with the potential for well-intentioned but outdated advice to perpetuate unsafe infant care practice. With the maturation of identity comes agency [51]; however, what these data revealed was more akin to the adolescent–parent ‘battle of will’ rather than agency. The new grandparents perhaps think they know best and have had experience of parenting. The young mother has no experience and may still be viewed as a child who is unable to make her own decisions or the right decisions, which possibly relates to ‘teen brain’ functioning. In this ‘battle of will’, the new grandparent may push their point, and the young mother may capitulate when overtired and stressed by her new and unfamiliar role, not having the correct information, or when lacking the confidence to confront her parent. There may also be a mistrust of professionals with new and different advice to family experience, which can result in guidance being rejected [30]. Smithbattle [30] describes these situations as ‘family legacies’; this culture within the family can have a significant influence on the young mother’s ability to embed recommended infant-care practices and risk-reduction measures into her own infant care. Therefore, the impact of relationships within the family is significant, particularly when the young mother is living in the family home.

Once mothers, all participants talked of rapid and inevitable change in terms of them having to grow up quickly, take control of their life, and make important decisions for both themselves and their babies, findings also identified by Leese [5] and Rolf [23]. Participants negotiated changes in their identity, relationships, and the external pressures and expectation that they should move toward more adult decision making and behaviour.

Challenges regarding identity and relationships were a recurring theme for participants. Erikson [45] describes the requirement for sequential progress through eight developmental stages, each presenting a psychosocial ‘crisis’ to be resolved to construct an individual’s personality. Stage five states that finding a sense of self and developing a personal identity are critical tasks during adolescence, and failure to achieve this can result in role confusion and a lack of understanding of self and one’s role in society and relationships. This negotiation of identity crisis was evident for the younger participants; they were still working out who they were and appeared ‘lost’ and isolated from their peer group, without ‘real friends’. Although they could state future goals and were aspirational, they were defining their identity in terms of motherhood and adult responsibilities and regretted not knowing who they would have become if they had not got pregnant [5,23]. Older participants, by contrast, had a clearer understanding of who they were and where they fitted in society.

Adolescent brain development involves significant remodelling of neural pathways in the ‘thinking’ and ‘processing’ areas in the pre-frontal cortex (PFC). The PFC controls executive functioning and is concerned with decision making, problem-solving, future planning, understanding consequences, and impulse control; this remodelling can take many years to complete. In the interim, adolescents may rely on solving problems and making decisions on a different area of the brain, the amygdala, which is associated with more emotional, impulsive, and instinctive or reactive behaviour [42,52]. Although adolescents are developing the ability to make rational and considered decisions [53,54,55], some of their decisions will still be based on more emotional processes and, therefore, appear (or are) illogical, impulsive, or reactive [42,44,56]. Evidence of this vacillation in decision making was captured in the findings of this study; unlike the average adolescent, for these young mothers, the consequences of some of their less-considered decisions, having their infant in bed to sleep with them for example, could have a tragic outcome. More often, decisions lacked explicit, complete, or accurate rationale, or participants were ‘instructed’ by someone they trusted and did not question or challenge the basis of the instruction, findings shared by Caraballo [11] and Pease [12]. This lack of clear understanding leaves room for the perpetuation of unsafe practices through using outdated or adapted advice, and a failure to apply risk assessment consistently to their parenting activity.

Revisiting the themes identified in the literature, participants in this study reported similar experiences, both positive and negative, to young mothers in previous research. They were not particularly well equipped or prepared for motherhood but tried to do the best they could with the resources they had. However, this was influenced by external factors such as family, interactions with professionals, a lack of resources and knowledge, and their functioning as adolescents. Participants were aware of the negative stereotypes and reported this impacted their experience of pregnancy and motherhood; public perception was experienced largely as judgement and derision and undermined their confidence and self-esteem but was also experienced within some family and professional relationships. Occasionally young mothers were undermined by their parents and professionals and grew accustomed to expecting to be judged, which caused conflict within family relationships and alienated them from universal services.

Pregnancy and motherhood did, however, impact them positively, demonstrating some positive behaviour change and aspirations for the future. They just didn’t always have the necessary support and information. Participants expressed wanting to be taken seriously as mothers and to actively participate in discussions of relevant topics related to infant care practices. They valued explanation, time to process information, and help to challenge outdated advice, although they struggled to ask for help, fearing they would reveal themselves to be inadequate mothers or that such help seeking would evoke ‘being told’ what to do.

These findings have implications and relevance for professionals providing services to and aiming to engage effectively with this group of mothers. Professionals should consider the requirements of adolescence in strategies to engage with them; however, even within the age range of these participants, 16 to 19 years, there were clear differences between older and younger participants in terms of ‘readiness’ for motherhood, but significant similarities in their approaches to decision making and behaviour. Interestingly, the younger participants wanted more information and explanation, wanting to learn how to be good mothers, and as such are potentially easier to engage than the older mothers who felt they already knew how to care for their infant and did not need any advice, preferring to work things out themselves. Significantly, there was disruption and adaptation of identity and relationships for all participants during pregnancy and early motherhood. While the younger participants attempted to overcome these challenges individually, specialist and relevant service provisions could have supported this activity. Erosion of specialist service provision impacts on opportunities for these young women to socialise and share experiences with other young mothers and curtails access to professional support and current health promotion and education opportunities [57,58,59]. Fishbein and Ajzan [60] argue that external social acceptance of behaviours as the ‘norm’ and behaviour associated with peer group activity and identity have a significant influence on an individual’s motivation and ability to engage with behaviour change. Although both younger participants were in receipt of the FNP service, this provision is delivered by individual home visits and does not alleviate the isolation that young mothers might experience; this was illustrated by both younger participants expressing that they felt ‘trapped in their bedrooms’, and simple activities such as going out were now like ‘expeditions’ but neither did they have anywhere to go.

The younger participants were also more likely to describe how hard the mothering role was. For young mothers, still living in the family home can present its own set of challenges for both mothers and professionals. It is important to engage the young mother and treat them with respect, but it may be advantageous to include the new grandparent which must be achieved without undermining the young mother; the outcome is critical to the ongoing influence the professional has in imparting knowledge and information and influencing change.

Data from this study highlight that adolescent brain functioning influences these young mothers’ parenting decision making and resultant behaviour. Young mothers who continue to live in the family home must negotiate help and advice from a range of sources, including potentially overbearing family members and judgmental professionals. Of note, participants both wanted advice and information, but often rejected it, depending on how, and in what context, it was delivered. Professionals need to take account of the functioning and impact of this important developmental stage and its characteristics to develop effective strategies to build trusting and respectful relationships with this high-risk group of mothers. Services designed with this in mind might positively influence the adolescent’s experience in transitioning to motherhood and engage them more effectively in health promotion and education opportunities to improve outcomes for both young mothers and their infants.

Recommendations from this study:Professionals should consider the impact of adolescent development on young mothers’ ability to engage with and interpret health promotion messagesHealth promotion messages delivered with a more ‘coaching and conversational’ approach may be more effective for young mothersTargeted and age-appropriate services such as breastfeeding cafes can offer a more appropriate and inviting environment to engage young mothers with health promotion and education opportunities

The strengths of this study are that the voice of the young mother was represented in the richness and depth of data collected across three timepoints during the ante-and post-natal periods (the greatest risk period for SIDS to occur) and covered all seasons. Interviews also offered the opportunity to consider each infant’s sleep environment and parent–infant interaction. The limitations of this research are that this was a small qualitative study with five participants in one English city and represents the individual experiences of a homogeneous group of participants. While these findings cannot be transferred to the population of young mothers per se, significant similarities were shared in terms of the experiences participants reported, providing an insight into the lived experience of young mothers not previously described in the literature. Further interpretation and transferability of these concepts should be treated with caution and are for the reader to consider. However, this study highlights significant new insights to be explored in studies with different participant groups, for example, fathers, grandparents, and professionals.

## 5. Conclusions

The participants in this study were, first and foremost, adolescents. This key developmental stage was superimposed with the transition to motherhood; however, these young women did not become adults when they became mothers. Adolescence, therefore, was the context which influenced their ability to engage with health promotion, which, in turn, influenced their decision making, behaviour, and infant-care practices. Data from this study highlight the importance of understanding how health promotion topics shared with young mothers are received and processed, and ultimately how that knowledge is translated and applied as safer parenting practices. This research provides a deeper understanding of the lived experience of young mothers, and the findings shared from this theme may support professionals to engage more effectively with this high-risk group of mothers and contribute to developing more effective and relevant interventions to improve uptake of safer sleep and parenting messages in this group and reduce infant deaths.

## Figures and Tables

**Figure 1 children-10-00904-f001:**
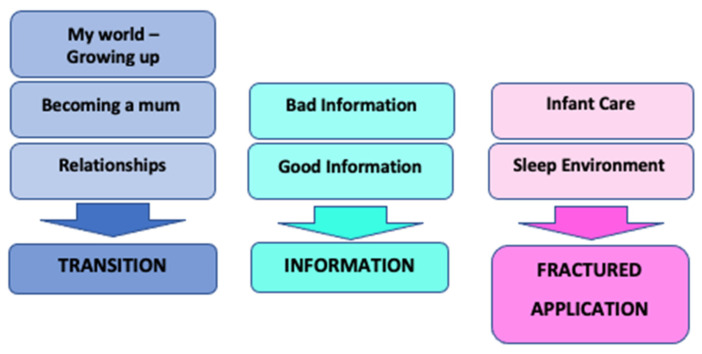
Superordinate Themes.

**Figure 2 children-10-00904-f002:**
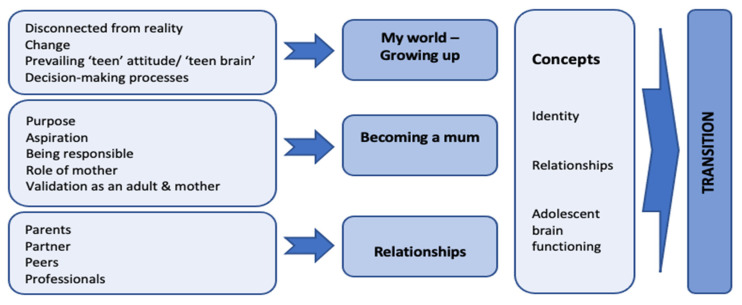
Transition: Emergent Themes.

**Table 1 children-10-00904-t001:** Participant profiles.

Pseudonym	Age	Father	Occupation	Income	Housing
Mia	16	No relationship/No contact	School	Nil	Family home
Sophia	16	No relationship/Some contact	School	Nil	Family home
Grace	17	Relationship status unclear	Unemployed since school	Some benefits	Council flat but remained in family home
Ruby	18	Living with	Working—apprenticeship	Maternity allowance	Family home with partner
Evie	19	Relationship/living apart	Unemployed since school	Nil	Family home

## Data Availability

Published in Ph.D. Thesis: Ellis C. Safely Sleeping? An Exploration of Mothers’ Understanding of Safe Sleep Practices and Factors that Influence Reducing Risks in Their Infant’s Sleep Environment. (Doctoral Thesis), University of Warwick. (2019).

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
