# Peer review of "Adolescence as the Context for Understanding Young Mothers’ Engagement with Health Promotion: A Phenomenological Exploration†"

_children, 2023, doi:10.3390/children10050904_

Round 1

Reviewer 1 Report (Previous Reviewer 1)

Dear authors, I have read your improved version with interest. I think that the methods could still be better sent, and the interpretation of the results could be deeper. However, I consider the overall scope of the text under consideration.

Author Response

Dear Reviewer,

Thank you for your consideration of this work.

Please can I ask what further information you wish me to include regarding the methodology section? A specific steer might help me understand what is missing.

Thank you.

Reviewer 2 Report (New Reviewer)

This paper presents data from a small qualitative study examining the lived experience of five young women transitioning to motherhood using longitudinal phenomenological analysis.

The title and abstract within the manuscript differ from the abstract submitted online. The latter needs to be updated.

I also see that revisions appear to be highlighted for this manuscript but I am unable to see any previous reviewers’ comments so it is unclear to me what the revisions are responding to. 

The majority of the references cited, particularly those relating to public health and SIDS-risk issues are somewhat old (pre 2018). Are there any newer citations that could be included? 

In the final paragraph of the methods section the 1st person is introduced when discussing Yardley’s framework. This is out of keeping with the remainder of the methods which are not presented in the first person, so this should be revised.

It seems somewhat confusing to introduce all 3 superordinate themes in the findings but to discuss only one. I would delete the initial paragraphs of ‘Findings’ and Figure 1 and focus exclusively on discussing the theme of transition, noting that other themes will be discussed in subsequent papers. This would make this paper more relevant to a range of public health topics where the transition to motherhood during adolescence may be relevant.

In the section on ‘Growing up’ it is noted that: ‘Interestingly, the parents were equally complicit in this, and were less likely to view their daughters as moving toward adulthood.’ It would be helpful to provide some explanation as to the source of this information? Did the authors interview parents of the teenage mothers who participated in this study, or was this information gleaned through the lens of the teen mothers themselves?

Last para of page 6 Is rather convoluted with multiple ‘or’ statements. Suggest rewriting for clarity.

Likewise second paragraph of discussion is long, convoluted with multiple clauses – rewrite.

The term ‘The young mum’ is used within the discussion – this seems too informal unless part of a quote.

There is a useful paper by Volpe et al (2014) presenting case studies of adolescent and adult mothers’ night-time infant care practices in a US sleep lab study. This may provide some helpful examples for reinforcing some of the concepts highlighted here. https://www.sciencedirect.com/science/article/pii/S0277953612005096?via%3Dihub

There are some grammatical issues that need addressing, particularly run-on sentences.

Author Response

Dear Reviewer, 

Thank you for taking the time to review my work. Please see my responses:

Feedback from first review round:

Title was changed to 'health promotion' to reflect the broader topic in the paper from the first reviews, however did not update when the paper was resubmitted into the system

The highlighted sections were the responses to feedback from the original reviews, which asked that the broader study themes be included for context to the theme of transition. I have moved this to the end of methodology section, as it does explain the 'relevance' of transition to the other themes. However, if you think it should removed, happy to do so.

References for SIDS were from seminal work presenting the background for the study. In subsequent papers I will be discussing more recent work reflecting relevance to the themes of information and application of safer sleep advice, which also includes ref to Volpe et al, which I consider to be relevant in the information and application theme papers which will follow this first publication.

Growing-up section: Inclusion of comments from 'grandparents' has been explained.

Grammatical, terminology and 1st person issues addressed.

This manuscript is a resubmission of an earlier submission. The following is a list of the peer review reports and author responses from that submission.

Round 1

Reviewer 1 Report

Thank you very much for the opportunity to read your paper. I have a few recommendations for the final version of the paper:
1. The theoretical background and assumptions are absolutely not in line with the rest of the text (the SIDS issue, which is the subject of the introduction, needs to be dealt with in the rest of the paper).
2. A better and more accurate description of the research population and the research context would be helpful. Given its size, it is not possible to infer the validity of the results without describing the characteristics of the participants and the context in which the research was conducted.
3. In the conclusion or discussion, it would be helpful to specify the recommendations.

Reviewer 2 Report

This is a narrative study of young adolescent mothers. The study examines attitudes of the young women to their motherhood. The study sets out to explore safe sleeping environments for the young mother's babies but as far as I can tell no specific questions addressing this was put. For instance, were the mothers aware of risky sleeping surfaces, (parental bed, sofa, used mattresses), etc. It would have been helpful if some idea of the mothers' understanding of why these surfaces are potentially risky was presented. Breastfeeding was addressed but the young mothers were never asked about breastfeeding and SIDS, or why it is advantageous in this regard.  It would improve the study if these issues were included.

Reviewer 3 Report

Dear Authors,

Thank you for inviting me to review this manuscript. The paper address a very important topic for women's health.

The paper itself is well written.

The authors undertaken a rigorous piece of data collection and have analyze information accurately. Moreover the authors conducted a thorough  literature review.

It was a pleasure to read this manuscript.

With best and warmest regards,